# The importance of the interface for picosecond spin pumping in antiferromagnet-heavy metal heterostructures

Farhan Nur Kholid[1], Dominik Hamara[1], Ahmad Faisal Bin Hamdan[1], Guillermo Nava Antonio[1], Richard Bowen[1], Dorothée Petit[1], Russell Cowburn[1], Roman V. Pisarev [2], Davide Bossini [3], Joseph Barker [4] ✉ & Chiara Ciccarelli [1] ✉

Interfaces in heavy metal (HM) - antiferromagnetic insulator (AFI) heterostructures have recently become highly investigated and debated systems in the effort to create spintronic devices that function at terahertz frequencies. Such heterostructures have great technological potential because AFIs can generate sub-picosecond spin currents which the HMs can convert into charge signals. In this work we demonstrate an optically induced picosecond spin transfer at the interface between AFIs and Pt using time-domain THz emission spectroscopy. We select two antiferromagnets in the same family of fluoride cubic perovskites, $KCoF_3$ and $KNiF_3$, whose magnon frequencies at the centre of the Brillouin zone differ by an order of magnitude. By studying their behaviour with temperature, we correlate changes in the spin transfer efficiency across the interface to the opening of a gap in the magnon density of states below the Néel temperature. Our observations are reproduced in a model based on the spin exchange between the localized electrons in the antiferromagnet and the free electrons in Pt. Through this comparative study of selected materials, we are able to shine light on the microscopy of spin transfer at picosecond timescales between antiferromagnets and heavy metals and identify a key figure of merit for its efficiency: the magnon gap. Our results are important for progressing in the fundamental understanding of the highly discussed physics of the HM/AFI interfaces, which is the necessary cornerstone for the designing of femtosecond antiferromagnetic spintronics devices with optimized characteristics.

Antiferromagnetic spintronics aims to harness the advantages of THz-frequency magnetic dynamics and low sensitivity to magnetic fields to develop new types of memory and logic devices[1–4]. While the spin transport properties of antiferromagnets have been extensively investigated with electrical methods[5–14], the high eigenfrequencies of the antiferromagnetic resonances suggest that femtosecond laser pulses may be the most promising tool to fully explore the potential of antiferromagnetic spintronics. The all-optical generation, manipulation and detection of magnons in bulk antiferromagnets have been widely demonstrated[3,15–18] up to the frequency regime of tens of

[1]Cavendish Laboratory, University of Cambridge, Cambridge CB3 0HE, UK. [2]Ioffe Institute, Russian Academy of Sciences, 194021 St. Petersburg, Russia. [3]Department of Physics and Center for Applied Photonics, University of Konstanz, D-78457, Konstanz, Germany. [4]School of Physics and Astronomy, University of Leeds, Leeds LS2 9JT, UK. ✉e-mail: j.barker@leeds.ac.uk; cc538@cam.ac.uk

**Fig. 1 | Terahertz emission spectroscopy in AFI-Pt. a** Schematic illustration of the measurement layouts. In Layout 1 the 800 nm pump impinges on the sample from the Pt side and the THz emission is measured behind the antiferromagnetic insulator (AFI). In Layout 2 the 800 nm pump impinges on the sample on the AFI side while the emitted THz pulse is measured behind the Pt. The external magnetic field is shown by a blue arrow and is directed along the [100] direction. **b** Crystal and magnetic structure of $KCoF_3$ and $KNiF_3$. **c, d** Magnetic susceptibility and its first derivative with respect to temperature in $KCoF_3$ (**c**) and $KNiF_3$ (**d**). From these datasets we extract the Néel temperature $T_N = 117$ K in $KCoF_3$ and $T_N = 245$ K in $KNiF_3$.

$THz$[19,20] and even in the absence of lattice heating[21]. As a next step towards establishing femtosecond antiferromagnetic devices, it is paramount to identify pathways to establish an efficient photo-induced spin-charge conversion. Recently, there have been a large number of studies on the conversion of high frequency spin excitations into charge signals in AFI/HM interfaces[12,13,22,23]. Some of these works have sparked controversy, because contrasting results have been reported by different groups in very similar structures[22–24]. Hence questions concerning the spin-to-charge conversion mechanisms and the role of AFI/HM interfaces on the femtosecond timescale have arisen.

Here we focus on a yet-unexplored route to transfer spin angular momentum from an AFI to a HM on the ultrashort timescale: the picosecond spin Seebeck effect (SSE). The quasi-static SSE has been measured in different AFIs[25–27] revealing a rich and not fully understood phenomenology. Satisfactory modeling of the results obtained in the presence of a magnetic field below the spin flop value, i.e. for vanishing net magnetization, has been obtained by extending the theory of bulk magnon transport developed for ferromagnets and taking into account the contribution from the two magnon branches carrying opposite spin momentum[28]. These branches degenerate at zero magnetic field: an applied field thus lifts the degeneracy and enables net spin transport. In this scenario, the SSE amplitude is largely affected by the spin transport properties in the bulk of the AFI. Nevertheless, recent experiments as a function of temperature and magnetic field[27,29] have pointed towards the key role of the interface[30]. It is clear that lacking an understanding of the spin transfer mechanisms between the AFI and the HM prevents to achieve a complete modeling and understanding of the experiments.

Our study here addresses exclusively the role of the interface and characterises the spin transfer efficiency between an AFI and a HM as a function of temperature and applied magnetic field. We focus on the two interfaces in $KNiF_3$/Pt and $KCoF_3$/Pt. Both AFIs have a simple cubic perovskite-type structure and their spin structures have the same magnetic space group. However, $KNiF_3$ has a lower magnetic

anisotropy, resulting in a magnon gap that is about an order of magnitude lower with respect to $KCoF_3$.

In our study, we use an optical pump-THz emission layout illustrated in Fig. 1a. Femtosecond laser pulses excite the free electrons in the HM thus raising their temperature. At the initial stage of the process, the excited hot electrons are out of thermal equilibrium with the magnons and the crystal lattice. The subsequent thermalisation mechanisms lead to the transfer of a spin current from the AFI to the Pt, where it is converted into a charge current due to the inverse spin Hall effect (ISHE). The narrow bandwidth of our measurements (0.25–2.5 THz) filters any contribution to spin pumping from thermal magnons or phonons. Hence our measurements are sensitive only to the direct electron-magnon interaction.

## Results

### Model for the optically induced incoherent spin pumping at an AFI-Pt interface

The onset of long-range magnetic ordering affects the nature of spin transport across a magnetic insulator (MI)-HM interface. Phenomenologically, interfacial spin transport can be understood in terms of incoherent spin pumping. The photo-induced thermal fluctuations of itinerant electron spins in the HM generate torques on the adjacent localized spin moments of the MI and affect the magnon population in the vicinity of the interface. Vice versa, the scattering of magnons at the interface leads to spin accumulation in the HM. The coupling strength between the itinerant spins in the HM and the localized spins of d-electrons in the MI is often parameterized by an sd-exchange Hamiltonian with constant $J_{sd}$. In thermal equilibrium, scattering processes from both sides of the interface are equal in magnitude, resulting in a vanishing spin current. An imbalance between the magnon and the electron distributions is therefore required to break the equilibrium and to yield a net spin flow.

In ferrimagnetic insulators, the magnon modes of opposite polarization are separated due to the inter-sublattice exchange field[31]. In uniaxial AFIs, the two magnon modes with opposite polarizations are

degenerate at zero magnetic field. Hence even when thermal equilibrium is broken, the equal flow of spin current with opposite sign from the two magnon branches results in zero net spin flow. Breaking the degeneracy via Zeeman splitting is therefore necessary to cause an unequal population of the two magnon branches and lead to a net spin flow. As well as the mode degeneracy, AFIs can have a large zero-field magnon gap in the THz range due to the 'exchange enhancement' of the anisotropy. This is significantly larger than the GHz gap of ferromagnets and can be expected to affect spin transfer across AFI-HM interfaces.

To understand how the relevant material parameters modify the optically-induced, incoherent spin pumping at picosecond timescales, we combine a Heisenberg model for the AFI, a Hubbard model for the HM and an sd-Hamiltonian coupling them at the interface, $\mathcal{H} = \mathcal{H}_{AFI} + \mathcal{H}_{HM} + \mathcal{H}_{sd}$. This follows a similar approach used to study picosecond demagnetisation dynamics in bulk metallic ferromagnets[32]. The sd-coupling is treated as a weak term in the Hamiltonian allowing a solution in the mean field with Fermi's Golden rule. A full derivation is given in the Supplementary Information. Here we outline the basic equations. Diagonalising the Hamiltonian and applying Fermi's Golden rule we find the spin current per unit area across the AFI-HM interface to be

$$
\begin{aligned}
I_{sd} = \pi S J_{sd}^2 (D^{E_F})^2 a_{AFI}^3 a_{NM}^4 \Bigg\{ & \int_{\varepsilon_0^\alpha}^{\varepsilon_{max}^\alpha} d\varepsilon_q^\alpha g_\alpha\left(\varepsilon_q^\alpha\right) \left[N_q + U_q\right] \left[\varepsilon_q^\alpha + \mu_s\right] \left[n_B\left(\varepsilon_q^\alpha + \mu_s\right) - n_\alpha\left(\varepsilon_q^\alpha\right)\right] \\
& - \int_{\varepsilon_0^\beta}^{\varepsilon_{max}^\beta} d\varepsilon_q^\beta g_\beta\left(\varepsilon_q^\beta\right) \left[N_q + U_q\right] \left[\varepsilon_q^\beta - \mu_s\right] \left[n_B\left(\varepsilon_q^\beta - \mu_s\right) - n_\beta\left(\varepsilon_q^\beta\right)\right] \Bigg\}
\end{aligned}
$$

(1)

where $S$ is the absolute value of the AFI spins in units of $\hbar$, $a_{AFI}$ and $a_{NM}$ are the AFI and NM lattice constants. $\alpha$ and $\beta$ denote the two AFI magnon modes (degenerate in zero field) with dispersion $\varepsilon_q^{\alpha,\beta}$. $\varepsilon_0^{\alpha,\beta}$ are the energies at the bottom of the magnon bands and $\varepsilon_{max}^{\alpha,\beta}$ are the energy maxima at the Brillouin zone edges. $g_{\alpha,\beta}(\varepsilon_q^{\alpha,\beta})$ are the magnon density of states (DOS) of the involved modes and $D^{E_F}$ is electron DOS at the Fermi level. The terms $N_q + U_q$ represent the normal and umklapp scattering, respectively. The thermal distribution $n_B(\varepsilon) = (\exp(\varepsilon/k_B T_e) - 1)^{-1}$ is the Bose–Einstein distribution function for the electron-hole pairs at the effective electron temperature $T_e$, whilst $n_{\alpha,\beta}(\varepsilon)$ are the instantaneous (non-equilibrium) distribution of magnons.

When the femtosecond laser pulse hits the sample, the electronic temperature in the HM increases rapidly, however the magnon population in the AFI is initially still at the ambient temperature. This temperature bias, or more generally, the difference between the distributions $n_B\left(\varepsilon_q^{\alpha,\beta} \pm \mu_s\right)$ and $n_\alpha\left(\varepsilon_q^{\alpha,\beta}\right)$ gives rise to a spin current across the interface.

The magnon thermal distribution $n_{\alpha,\beta}(\varepsilon)$ evolves in time due to the spin current

$$
\frac{\partial n_{\alpha,\beta}\left(\varepsilon_q^{\alpha,\beta}\right)}{\partial t} = \frac{I_{sd}\left(\varepsilon_q^{\alpha,\beta}\right)}{\hbar}
$$

(2)

We can assume that magnon-magnon scattering within and between magnon branches is negligible on the short timescales studied here. The contribution to the spin current from the $\alpha$ and $\beta$ modes differs in sign since the both modes pump spins with the opposite signs due to their opposite magnon polarization. The spin current flow induces a spin accumulation $\mu_s$ in the HM according to an equation

$$
\frac{\partial \mu_s}{\partial t} = -\frac{\mu_s}{\tau_s} + \frac{\rho}{\hbar} I_{sd},
$$

(3)

where $\tau_s$ is the relaxation time of s-electron spins in the HM and $\rho = -1/D$ with $D = 2D_\uparrow^{E_F} D_\downarrow^{E_F}/(D_\uparrow^{E_F} + D_\downarrow^{E_F})$. This spin accumulation

further influences the spin current (Eq. (1)) because the contribution from the $\alpha$ mode is increased whilst that of the $\beta$ mode is decreased. This process is similar to the effect of an applied field on the AFI dispersion. Hence the lower energy magnon mode has an increasingly dominant effect with increasing of $\mu_s$.

The net spin current is transient. As the magnon distribution in the AFI equilibrates towards the thermal distribution of the electron-hole pairs in the HM, the spin accumulation decays to zero and the net spin current once again vanishes.

There are competing processes taking place at different timescales that define the temporal evolution of both the effective electronic temperature and the spin accumulation. This is mostly determined by the strength of the interface coupling through $J_{sd}$, $\tau_s$ and $\rho$ parameters. When the sd coupling is stronger, the magnon distribution follows the electron-hole distribution more closely, but rapidly creates a significant spin accumulation which feeds back into enhancing the spin current. When the sd coupling is weak, the electron temperature reaches a maximum value and then decreases before the magnon distribution has changed significantly. Hence less spin accumulation is created, resulting in the reduction of the spin current that can be generated.

The magnon dispersion also affects the magnitude of the spin current generated by the laser-pulse heating. The enhancement of the magnetic anisotropy in antiferromagnets results in significant increase of the magnon gap at low wavevectors in comparison to ferromagnets. The gap is approximately $\varepsilon_0^{\alpha,\beta} \approx 2S\sqrt{(zJ_{AFI} + K)K} \mp g\mu_B\mu_0 H_{ext}$, where $J_{AFI}$ and $K$ are the exchange and anisotropy constants respectively, and $z$ is the number of nearest neighbours. The second term is the Zeeman energy in the presence of an external magnetic field $\mu_0 H_{ext}$. The magnon gap defines the forbidden energy range within which there are no magnon states for electrons to scatter into, hence the magnitude of the spin current at low temperatures is limited by this gap. The additional heating of electrons by the laser-pulse has a diminishing effect when approaching the Néel temperature ($T_N$) because of the absence of any magnon modes beyond the magnon Debye temperature. The spin current amplitude therefore saturates when approaching $T_N$.

**Experimental Layout**

In this work, we compare the optically induced incoherent spin pumping at the AFI-HM interface in $KCoF_3$ and $KNiF_3$ bulk single crystals (see Methods). Below $T_N$ both materials are G-type antiferromagnets, with neighbouring spins aligned along opposite directions and pointing along one of the main cubic [100], [010] and [001] axes[21] (Fig. 1b). $KNiF_3$ is often regarded as an almost perfect Heisenberg antiferromagnet because of its low magnetocrystalline anisotropy, which produces resonance frequency $\omega_{KNiF_3} = 0.097$ THz[21] at zero field. In $KCoF_3$, the unquenched orbital angular momentum of the $Co^{2+}$ ions results in a larger spin-orbit interaction, which is responsible for a larger anisotropy. Consequently, the resonance frequency $\omega_{KCoF_3} = 1.17$ THz[33,34] at zero field is higher by an order of magnitude than that of $KNiF_3$. Fig. 1(c) and (d) show the temperature dependence of the magnetic susceptibility for the two materials, from which we extract the values of the Néel temperature $T_N = 117$ K in $KCoF_3$ and $T_N = 245$ K in $KNiF_3$.

We use 50 fs pump laser pulses with a central wavelength of 800 nm to generate a temperature difference between the free electrons in Pt and the magnons in the AFI. Fig. 1a shows the experimental layouts: in Layout 1 the pump impinges on the sample from the Pt side and the THz emission is measured behind the AFI. In Layout 2 the pump hits the sample on the AFI side while the emitted THz pulse is measured behind the Pt. Unless otherwise stated, in order to maximize the signal-to-noise ratio we used Layout 2 in $KCoF_3$/Pt and Layout 1 in $KNiF_3$/Pt. This choice is motivated by a transparency window around 800 nm for $KCoF_3$. However, this spectral range coincides

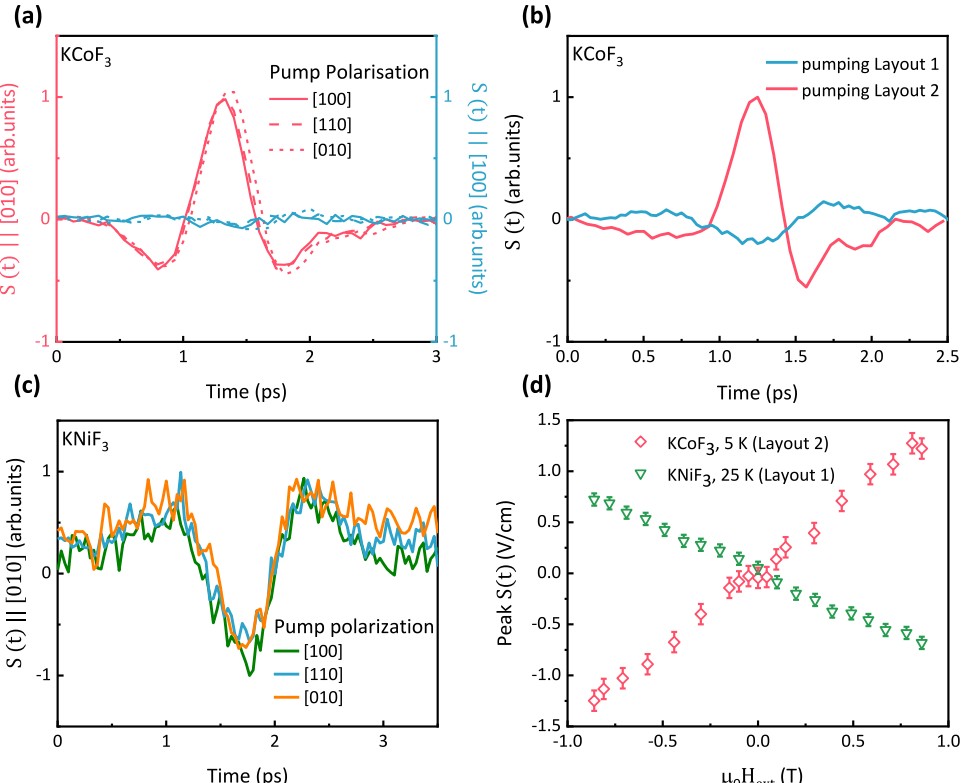

**Fig. 2 | Symmetry of the terahertz emission. a** Components of the THz emission from KCoF$_3$/Pt polarized along the [100] and [010] axes for different optical pump polarizations. The absorbed pump fluence is 1.7 mJ.cm$^{-2}$ the ambient temperature is $T = 20$ K and $H = +0.85$ T along the [100] axis. **b** THz emission for two pumping geometries in KCoF$_3$/Pt at $T = 20$ K and $H = +0.85$ T. The different optical paths in the two pumping layouts cause a relative delay-shift, which we have rescaled so that the two time-traces share the same time-axis and can thus be directly compared. **c** Pump polarization dependence of the THz emission in KNiF$_3$/Pt, measured at $T = 25$ K. **d** THz electric field emission in KCoF$_3$/Pt (red) and KNiF$_3$/Pt (green) as a function of applied magnetic field.

with a weakly-absorptive window with the absorption coefficient of $\alpha = 70$ cm$^{-1}$[21] for KNiF$_3$. Thus, in order to maximize absorption of the laser pulses by the Pt layer, in KNiF$_3$/Pt we need to pump directly on the Pt side and measure the emitted THz pulse behind the AFI.

After optical pumping, the hot electrons in the Pt layer thermalize with its phonon bath in Pt with a time constant of $\tau_{e-ph} \approx 260$ fs[35]. Subsequently, a Fermi-Dirac distribution at the effective temperature $T_e$ can describe the electronic system. At the timescales probed in our experiment, the thermalisation of the hot electrons in the Pt layer with the spins in the AFI mainly occurs via direct electron-magnon scattering mediated by the sd-type exchange $J_{sd}$. The typical time-scales of the phonon-magnon interaction in these AFIs, either directly measured with time-resolved techniques[21] or extrapolated via static spectroscopic methods[33], lie in the 10–100 ps region. We can thus assume that phonons do not significantly contribute to our measurements.

Although the hot electrons in the Pt layer are initially not spin-polarized, the incoherent magnons excited via inelastic spin-flip scattering lead to a non-equilibrium net spin accumulation in the AFI provided the degeneracy between the two opposite chirality bands is lifted by an external magnetic field. This results in spin pumping through the interface to the Pt layer where spin-to-charge conversion occurs via the inverse spin-Hall effect and results in broadband THz electro-dipole emission[36]

$$S(\omega) = eZ(\omega)\lambda_s \Theta_{SH} I_{sd}(\omega),  \qquad (4)$$

where $Z(\omega)$ is the impedance of the AFI-Pt bilayer, $\lambda_s$ and $\Theta_{SH}$ are respectively the spin diffusion length and the spin-Hall angle of the Pt layer, while $-e$ is the electron charge. The amplitude of the THz emission is directly proportional to the spin current $I_{sd}$ through the

AFI-Pt interface and therefore gives us a direct measurement of the optically-induced incoherent spin pumping.

## Analysis of terahertz emission

The emitted THz radiation is consistent with the symmetry of the inverse spin-Hall effect in Pt for a spin current polarized along the direction of the external magnetic field, which in our set-up is applied within the plane of the AFI-HM structure, as shown in Fig. 1a. Fig 2a, b show the time-trace evolution of the THz electric field pulse, $S(t)$, in KCoF$_3$/Pt at $T = 20$ K. The components polarized along the [100] and [010] directions are both shown for different linear polarizations of the pump beam (Fig. 2a). The THz electric field is polarized along the [010] direction, perpendicular to the direction of the external magnetic field. The pump polarization does not have any impact on the polarization or amplitude of the emission, in agreement with the thermal origin of the spin current. Figure 2b shows that the amplitude of the THz pulse is inverted when using the two reversed pumping layouts shown in Fig. 1a, i.e. direct illumination from either the KCoF$_3$ or the Pt side. This observation is consistent with the opposite direction of the spin current relative to the THz propagation direction, resulting in a reversal of the sign of the inverse spin Hall signal. In the Supplementary Information we show that THz emission is only measured when both the AFI, as spin source, and the Pt layer, as spin-to-charge transducer, are present in the structure while no emission was measured for the single AFI or a single Pt thin film deposited on glass. Similarly, in KNiF$_3$ the THz emission is polarized along the [010] axis and does not depend on the polarization of the pump beam (Fig. 2c). The profile of the emitted THz pulse appears reversed in KNiF$_3$ for the same magnetic field direction. This is because of the different geometry employed for the measurements of the KNiF$_3$/Pt heterostructure with the optical pump hitting the sample from the Pt side.

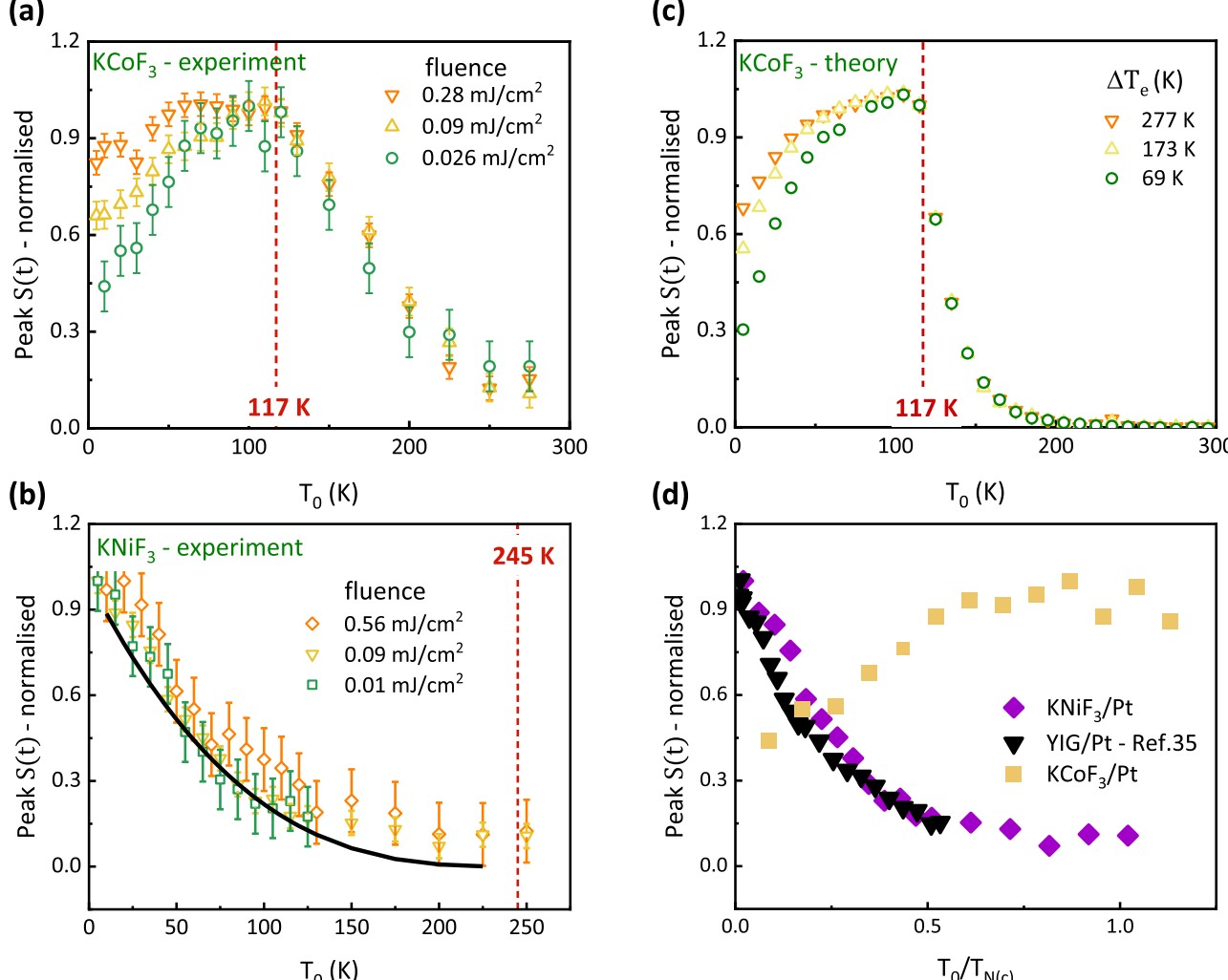

**Fig. 3 | Temperature dependence of terahertz emission. a** Temperature dependence of the peak THz emission in KCoF$_3$ for varying pump fluences. **b** Temperature dependence of the peak THz emission in KNiF$_3$ for varying pump fluences. The black line represents the function $\left(1 - \frac{T}{T_N}\right)^3$. Error bars represent the noise level in our time-domain THz emission data. **c** Calculated integrated spin current in KCoF$_3$ for different changes in the effective electron temperature in Pt with respect to the ambient temperature; $\triangle T_e = T_e - T_0$. The calculations have been done for a field applied along the [100] domain. **d** Temperature dependence of the THz emission amplitude in KNiF$_3$, KCoF$_3$ and YIG/Pt as presented in ref. [32]. as a function of the normalised ambient temperature.

In Fig. 2d we show that the amplitude of the emitted THz radiation linearly increases as a function of the external magnetic field in both the KCoF$_3$/Pt and KNiF$_3$/Pt structures. The opposite sign for the two samples is again explained with the different pumping geometries used: Layout 2 in KCoF$_3$/Pt and Layout 1 in KNiF$_3$/Pt.

**Temperature dependence**

Figure 3a, b show the normalised THz emission amplitude in the KCoF$_3$/Pt and KNiF$_3$/Pt samples as a function of temperature and for a constant value of the external magnetic field $\mu_o H_{ext} = 0.85$ T. For $T < T_N$ both AFIs manifest very different behaviour. In the KCoF$_3$/Pt sample the optically induced spin pumping peaks at $T_N$ and decreases when approaching zero. The temperature dependence is strongly affected by the pump fluence below $T_N$. This behaviour is reproduced by the theory described above by considering a gap in the magnon DOS of 1.17 THz (Fig. 3c). In the model we assume, that the laser-induced increase of the effective electronic temperature $\triangle T_e$ is independent from ambient temperature, as is found in transient reflectivity measurements[35]. The decrease of spin pumping below $T_N$ is explained by the gap $\varepsilon_0$ in the magnon DOS. After optical pumping, only those electrons in Pt that have sufficiently high energy above $\varepsilon_0$ can

contribute to the spin transfer by exciting high-energy magnons. At low ambient temperature the magnon gap $\varepsilon_0^{KCoF_3}$ in KCoF$_3$ becomes commensurate with the energy of the optically heated electrons in Pt, $k_B T_e$ ($\sim 6$ meV at an absorbed fluence of 0.1 mJ/cm$^2$)[35,37], which leads to the quenching of electron-magnon scattering, as illustrated in Fig. 4. As the fluence of the pump pulse is increased the electron temperature $T_e$ is proportionally increased and higher-energy magnons above the gap can be accessed, resulting in a larger value of the integral in Eq. (1).

In KNiF$_3$ the temperature dependence of the THz emission is different, increasing rather than decreasing towards lower temperatures (Fig. 3b). Moreover, no dependence on the fluence of the pump beam is observed: all curves overlap and are fitted by $\left(1 - \frac{T}{T_N}\right)^3$ function. These are strong indications that the opening of the gap in the magnon DOS in this case does not play the main role in determining the temperature dependence of the THz emission. At low ambient temperature the magnon gap $\varepsilon_0^{KNiF_3}$ in KNiF$_3$ is one order of magnitude smaller than the energy of the optically heated electrons in the Pt layer even at the lowest fluence that was used in our experiment. In Fig. 3d we compare the data in Fig. 3a, c) with the same data measured by our group in YIG-Pt[35,37] using the same experimental layout. The KNiF$_3$ and

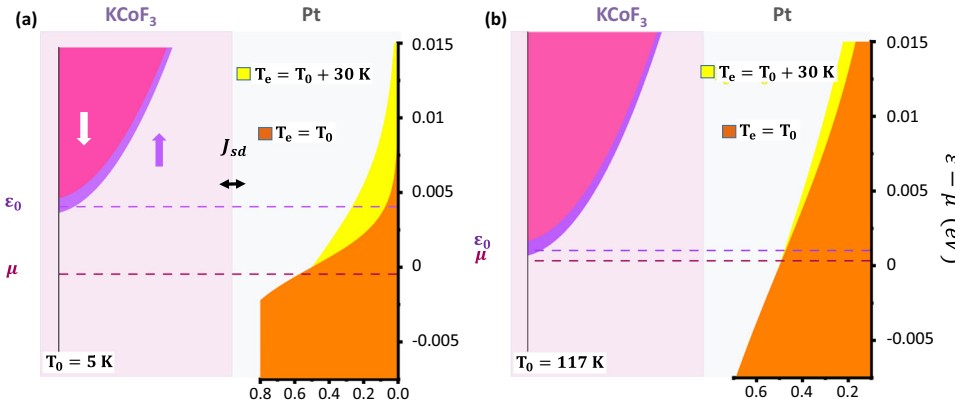

**Fig. 4 | Spin pumping across the AFI–Pt interface. a** At $T \ll T_N$, the gap in the magnon DOS fully opens subtracting the contribution of low energy magnons to the spin pumping. Increasing the laser fluence results in a higher effective electron temperature in Pt and higher energy magnons start to contribute to spin pumping. In the above graph we show the specific case of KCoF$_3$. The Fermi–Dirac distribution of the electrons on the Pt side is shown in orange at ambient temperature and in yellow after optical pumping (an electron temperature increase of 30 K is assumed). The two field-split magnon bands are shown in purple. **b** At $T \sim T_N$, the magnon gap is only partially open and low energy magnons are excited by spin-flip scattering by electrons close to the Fermi level in Pt.

YIG graphs overlap when plotted as a function of the renormalized temperatures $T/T_N$ and $T/T_c$, where $T_c$ is the Curie temperature in YIG. This suggests that the onset of the long-range magnetism and of the exchange coupling between the free electrons in Pt and the localized electrons in the magnetic insulator are responsible for the temperature behaviour. On the other hand, the role of quantities exclusively connected to Pt, such as the spin lifetime, do not seem to play a key role.

Finally, we briefly comment on the spin pumping in the paramagnetic state at $T > T_N$. In this temperature region there is no long-range magnetic order, but short-range spin correlations still exist and have been shown to lead to a non-zero SSE in both paramagnets[38] and antiferromagnets[27]. Within the model this is introduced phenomenologically by cutting off the spectrum starting from the centre of the Brillouin zone and for increasing wavenumbers with increasing temperature[39]. Assuming a correlation length of $\xi = a \left( \frac{T - T_N}{T_N} \right)^{-\nu}$ with $a = 0.3$ nm and $\nu = 0.64$ we cutoff the spectrum for wave vectors below $q_c = 1/\xi$. As shown in the Supplementary Information, the THz emission in the paramagnetic phase still reflects the symmetry of the inverse spin Hall effect in Pt and its amplitude scales linearly with the value of the external magnetic field. The temperature dependence in this case is described in terms of the critical scaling of the susceptibility in agreement with theory[30] and previous reports[27].

## Methods
### Material Characterisation
Single crystals KCoF$_3$ and KNiF$_3$ were grown using a solution-melt method technique. A 5 nm Pt layer was deposited on 0.8 mm thick plates of AFIs using magnetron sputtering after polishing the crystals plates to a surface roughness <10 nm. The crystals were polished using a Kent 3B automatic fibre optic polishing machine. Coarse polising of material by about 100 microns was done by using a 15 $\mu m$ diamond suspension on planacloth. Then, we gradually decreased the abrasive size from 15 $\mu m$ to 9, 6, and 3 $\mu m$. Between two stages, the sample was cleaned with de-ionized water to remove residues from the previous polishing stage. For the fine polishing 1 $\mu m$ of alumina silica was used on multicloth. De-ionized water is constantly added during the polishing to avoid the cloth getting dry. In the final stage, we polished the sample with 0.06 $\mu m$ alumina silica on multicloth and cleaned the sample with acetone and isopropanol. In addition to polishing, the surface was further cleaned by in-situ Ar-milling at the pressure of $2.2 \times 10^{-4}$ mbar (base pressure is $3.5 \times 10^{-8}$ mbar) before sputtering the Pt.

The perovskite crystal structure (Fig.1a) results in cubic anisotropy, where the Néel vector can point along one of the three [100], [010] and [001][40,41] crystal axes. Below $T_N$ KCoF$_3$ undergoes a tetragonal deformation ($\sim 0.2\%$ with c/a < 1), which results in a small uniaxial anisotropy with the easy axis along the [001] direction.

Using the SQUID magnetometry, we measured the Néel temperature of $T_N = 117$ K for KCoF$_3$ and of $T_N = 245$ K for KNiF$_3$. The low anisotropy results in large magnetic domain sizes in the range of hundreds of $\mu m$[40]. We refer the reader to the Supplementary Information for a more detailed discussion on domains.

### Experimental setup
800 nm femtosecond laser pulses with pulse width of 50 fs are generated from an amplified Ti:sapphire laser system with a repetition rate of 5 kHz. The laser spot on the sample surface is 2.5 mm in diameter. The nominal fluence of the incident pulse is varied from 0.04 to 3 mJ/cm$^2$, corresponding to an absorbed fluence by the Pt layer of 0.026 to 1.7 mJ/cm$^2$. Due to a low absorption of KCoF$_3$ at 800 nm ($\alpha = 10$ cm$^{-1}$) only $\lesssim 5\%$ of the total fluence is absorbed throughout entire thickness. On the other hand, a high absorbance in KNiF$_3$ of $\alpha = 70$ cm$^{-1}$ strongly reduces the laser fluence at the KNiF$_3$/Pt when incident from KNiF$_3$, and hence we prefer to pump from the Pt side. Constructive interferences in the thin Pt layer result in a high absorbance $A \approx \frac{n_i}{n_i + n_0} \approx 0.6$[42] for KCoF$_3$ ($n_i \approx 1.5$[43] and $n_0 = 1$ are the refraction indices of KCoF$_3$ and air, respectively) and $A \approx 0.4$ for KNiF$_3$. An electromagnet is used to apply an external field up to 0.85 Tesla. The resulting THz emission due to the inverse spin Hall effect is detected using an electro-optic sampling technique. The THz electric field radiation is steered by parabolic mirrors to a 1 mm thick ZnTe crystal such that it is spatially overlapped with a low power linearly polarized optical probe pulse (<1 nJ). The THz field-induced birefringence creates an ellipticity in the probe pulse that is proportional to the THz electric field strength. Balanced detection is employed to measure the small ellipticity changes and construct the THz signal in time-domain as the delay of the probe pulse is modified with a motorized linear stage. For signal conversion to electric field in units of V/cm unit, we use an equation[44]

$$\frac{\triangle I}{I_o} = \frac{\omega n^3 r_{14} L}{2c} 2S$$

where $\frac{\triangle I}{I_o}$ is the raw signal representing the change of the probe ellipticity, $\omega = 7.5\pi \times 10^{14}$ s$^{-1}$ is the central frequency of the probe pulse, $n = 2.85$ is the refractive index of the ZnTe detector at 800 nm, $r_{14} = 3.9$ pm/V is the electro-optic constant of ZnTe, $L = 1$ mm is the ZnTe thickness, $c = 3 \times 10^8$ m/s is the speed of light, $S$ is the THz electric field magnitude convoluted with the response function of our spectrometer[45,46].

## Model parameters

To model $KCoF_3$ we used the parameters $\mu_0 H_{ext} = 0.85$ T, $J_{AFI} = 0.66$ meV, $K = 0.4$ meV, $z = 6$, $S = 3/2$, $\tau_s(T) = 0.018 - 1.3 \times 10^{-5} T$ ps[44], $\tau_e = 0.26$ ps[35], $D^{E_F} = 0.015$ meV$^{-1}$ nm$^{-3}$ [40] $J_{sd} = 130$ meV[39] $a_{AFI} = 0.41$ nm, $a_{NM} = 0.39$ nm.

## Data availability

The datasets generated during and/or analysed during the current study are available in the Repository of the University of Cambridge Apollo at the address https://elements.admin.cam.ac.uk/viewobject.html?cid=1&id=1440949.

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

## Acknowledgements
C.C., D.H., R.B., and J.B. acknowledge support from the Royal Society. This project has received funding from the European Union's Horizon 2020 research and innovation programme under the Marie Skłodowska-Curie (grant agreement No. 861300) and the Engineering and Physical Sciences Research Council (grant number EP/V037935/1). F.N.K. acknowledge Jardine Foundation and Cambridge Trust for financial support. D.B. and C.C. are thankful to A. Kimel for the fruitful discussion and for sharing data taken in his group, which helped with the interpretation of the results presented in this paper. J.B. thanks Kei Yamamoto for advice on the theory of sd coupling with antiferromagnets. D.B. acknowledges support from the Deutsche Forschungsgemeinschaft (DFG) program BO 5074/1-1. RVP contribution to the paper was supported by Goszadanie.

## Author contributions
C.C. planned the experiment; F.N.K. and D.H. built the experimental set-up; F.N.K. took the measurements; F.N.K., D.H., G.N.A., D.B., J.B., C.C. contributed to the analysis and interpretation of the data; D.H., A.F.B.H., R.B., D.P. and R.C. fabricated the devices and characterised them with SQUID magnetometry, R.V.P. provided the single crystal antiferromagnets; D.B. did the pump-probe study of the antiferromagnetic modes; J.B. developed the theoretical model; F.N.K., D.H., A.F.B.H., G.N.A., R.B., D.P., R.C., R.V.P., D.B., J.B., and C.C. wrote the manuscript.

## Competing interests
The authors declare no competing interests.
