## [Peer Review File · Nature Communications]

Reviewers' Comments:

Reviewer #1:

Remarks to the Author:

Antiferromagnets, in comparison with ferromagnets, have attracted great attention recently because of their much higher resonance frequency, up to THz, and greater stability under external magnetic disturbance. Antiferromagnetic insulator (AFI) and heavy metal (HM) heterostructures have been frequently studied in the static electrical measurements, such as spin Seebeck effect (SSE) and spin pumping (SP), but there are only few studies for the ultrafast dynamics. In this manuscript, the authors report the THz emission in KNiF₃/Pt and KCoF₃/Pt. With the experiments by changing the polarization direction of pump laser-pulses, the incident direction, the magnitude of the external field and the temperature, the authors prove that the THz emission of KNiF₃/Pt and KCoF₃/Pt originates from the picosecond SSE, and the sd-coupling at the interface plays a key role. The different temperature dependent behaviors of these two samples are explained by the difference between the magnon gap of KNiF₃ and KCoF₃. The experimental results in this paper are relatively clean, and the experimental results are consistent with the physical mechanism given by the authors. The proposed mechanism gives deeper understanding on the picosecond spin transfer process at the interface of AFI/HM and the role of magnetic gap. Therefore, I suggest the publication of this manuscript after the authors address the following minor concerns.

1. There are three kinds of magnetic domains in the sample. When the magnetic field is applied along the [100] direction, the magnon modes in the domain whose Neel vector is along the [100] direction is different from those in the other two domains. All domains contribute to the THz signal. However, the contributions of different domains are not considered in the manuscript. If three domains are taken into account, the authors may make different conclusions.
2. It is more common to use Pt and W with opposite spin Hall angles to prove that the signal is from the inverse spin Hall effect. Therefore, it is suggested to replace Fig. 2b with the experiments on Pt and W. And in this fig, there should be differences in the positions of peaks for pumping from different side due to the delay of the photon propagation in the samples. However, I find that the THz signals peak at the same time.
3. The reason why the signals of KNiF₃ and KCoF₃ are opposite can be explained in more detail in the manuscript. It is not straightforward for the readers.
4. The temperature dependence is the key for this work. Do the authors consider the temperature rise caused by the thermal accumulation of the fs laser-pulses irradiation on the sample, particularly for using high frequency laser? How much impact will this have on the results of temperature dependence?

Reviewer #2:

Remarks to the Author:

The authors employ pump-probe THz emission spectroscopy in two different antiferromagnetic insulator-heavy metal (AFI/HM) heterostructures, namely KCoF₃/Pt and KNiF₃/Pt. The generated THz signal is attributed to spin current generation by heating the electronic system of Pt with the laser pulse and subsequent spin-charge conversion by the inverse spin Hall effect in Pt. When studying the THz signal in KCoF₃/Pt and KNiF₃/Pt as a function of temperature, the authors find qualitatively different temperature dependencies. For KCoF₃/Pt the THz signal decreases below the Néel temperature T_N and for KNiF₃/Pt the THz signal increases below T_N . Furthermore, the temperature evolution of the THz signal depends on laser fluence in KCoF₃/Pt but not in KNiF₃/Pt. The authors attribute these experimental findings to the different size of the spin-wave gap in these two antiferromagnetic materials. The observation in KCoF₃/Pt is in qualitative agreement with a theoretical model that employs the presumed large spin-wave gap (>1 THz) in KCoF₃. Spin transport in AFI/HM heterostructures is a relevant topic of current interest due to the rapidly evolving field of antiferromagnetic spintronics. Studies that probe these transport properties at ultrafast timescales are required to obtain a fundamental understanding of the underlying mechanisms.

I thus found the study intriguing and timely and it could in principle be suitable for publication in Nature Communications. However, I believe that more detail about the experimental procedure needs to be provided to make the work reproducible and some assumptions warrant additional

evidence / explanation. Specifically, these are my questions / requests:

- 1) The schematic illustration of the experimental setup in Fig. 1 (a) is insufficient. In the text the authors mention that they excite the bilayers either by illuminating from the AFI side or the Pt side. However, the exact geometry, such as the location of the THz detector relative to the sample and pump pulse is not explained clearly. The figure does not even label the AFI or HM, so for the reader it is unclear which geometry was used for the various datasets in Figs 2 & 3. To make the study reproducible please clearly state in the manuscript for each dataset what the measurement geometry was (e.g. Pump-> AFI/Pt -> Detector or Pump & Detector <-> Pt / AFI).
- 2) In Fig. 2 (b) was the change of illumination achieved by rotating the sample by 180°? Have the authors ensured that the orientation of the external magnetic field relative to the spin current polarization remained identical? Please explain the argument in line 198 that the relative propagation direction of the THz pulse and spin current changes the sign with a clear figure showing the assumed Néel vector / spin current flow direction / THz polarization.
- 3) As explained in the SI, the authors deal with a multi-domain state in KNiF3/Pt and according to Ref 3 in the SI the Néel vector of the dominant domains will be orthogonal to the external magnetic field. The Model (Eq. (1) in main text) assumes that the external magnetic field is aligned with the Néel vector. The model thus cannot be applied for KNiF3/Pt with a net magnetic moment stemming from the canting.
- 4) How do the authors ensure that they have a single domain state in KCoF3/Pt and that the external magnetic field is applied along the Néel vector?
- 5) I do not see a strong argument why the different size of the spin-wave gap should be the dominant (sole) cause of the observed different THz emission vs. temperature. Why is the magnitude of the net magnetic moment not relevant? In KNiF3/Pt, the net moment from canting (see point 3) would lead to a quasi-ferromagnetic spin Seebeck effect increasing with lowering T (as demonstrated by authors in Fig. 3 (d)) while in KCoF3/Pt (assumed as an ideal uniaxial AFM) the χ_{\parallel} decreases with decreasing temperature resulting in a reduced net moment for lower T (because the net moment stems from thermal fluctuations). A simple model that assumes that the spin-current generation is simply proportional to the net magnetic moment present at the given temperature and magnetic field then would seem to be sufficient to explain the observations. Why is the spin-wave gap model required/better/more plausible?
- 6) The fluence dependence below T_N is much stronger in experiment (Fig 3(a)) than in theory (Fig. 3(c)). Why?

Reviewer #3:

Remarks to the Author:

The authors describe their experiment and modeling of the generation of THz radiation by the so-called picosecond spin Seebeck effect created at an interface between a heavy metal and an insulating antiferromagnet in an applied magnetic field. The picosecond spin Seebeck effect was first identified in interfaces with insulating ferromagnets in 2017. This work extends the scope of these types of investigations to insulating antiferromagnets. The work is timely and important. I recommend publication in Nature Communications after the authors have considered the following criticisms:

- 1) The magnetic susceptibility of KCoF3 has a much weaker temperature dependence below T_N than KNiF3. The susceptibility of KNiFe3 behaves approximately as I would expect but KCoF3 does not. In fact, since the magnon gap is expected to be 1 THz, i.e., 4 meV, I would expect that the magnetic susceptibility would fall off more steeply with decreasing temperature than KNiF3 as the thermal excitations of magnons is quenched at low temperatures. The authors should discuss this point and provide a reasonable explanation of the observed magnetic susceptibility of KCoF3.
- 2) In Figure 2, the authors plot the data normalized by the largest value of "peak $S(t)$ " as a function of temperature. I find this practice misleading and it hides much of what I would like to compare in the data. I strongly recommend that the authors plot the observed (not normalized) "peak $S(t)$ " with the addition perhaps of dividing by the pump fluence when comparing data obtained at different pump fluences.
- 3) One of the questions I would like know the answer to (related to the point above) is the

magnitude of the signals for YIG/Pt and KNiF₃/Pt. The magnetization of KNiF₃ in the approximately 1 T field is orders of magnitude smaller than the magnetization of YIG. Is the THz generation also orders of magnitude smaller? If not, the authors should discuss the relative magnitudes of the signal for these two cases.

4) The authors must provide a thorough discussion of the preparation and surface characterization of the KCoF₃ and KNiF₃ crystals. In this version of the manuscript, all that is stated is that the crystals are polished to a roughness of 1 nm (which also seems to me to be extraordinarily small roughness for a mechanically polished crystal). Presumably the crystals are covered with adsorbed water vapor and hydrocarbons before loading into the deposition chamber. How are these surface contaminants removed prior to Pt deposition? This work is about interface physics and the structure and composition of the interfaces is important.

5) The authors state the values of the model parameters at the bottom of page 12 without discussing if these values are physically reasonable. For example, is it reasonable for J_{SD} to be 5 times larger than J_{AFI} ? The text states that J_{SD} is considered a small parameter yet its value is larger than J_{AFI} . The text states that the electron-phonon relaxation time in Pt is 0.2 ps but that value is given as 2 ps for the model.

6) At the end of the paper the authors state: "Our results set a landmark in the fundamental understanding...". I understand that many authors feel a need to hype their work to publish in so-called "high impact journals". The authors describe a thorough and important contribution to the literature of spin current generation and transport at interfaces. That is enough. Calling this work a "landmark" study is going too far.

Dear Dr. Bladwell,

We thank the reviewers for their comments and helpful input to improve our manuscript and we are grateful for the opportunity to reply. All remarks have been carefully considered and addressed point-by-point. Our response is attached below and a revised version of the manuscript is submitted for your consideration.

Reviewer 1

Antiferromagnets, in comparison with ferromagnets, have attracted great attention recently because of their much higher resonance frequency, up to THz, and greater stability under external magnetic disturbance. Antiferromagnetic insulator (AFI) and heavy metal (HM) heterostructures have been frequently studied in the static electrical measurements, such as spin Seebeck effect (SSE) and spin pumping (SP), but there are only few studies for the ultrafast dynamics. In this manuscript, the authors report the THz emission in KNiF₃/Pt and KCoF₃/Pt. With the experiments by changing the polarization direction of pump laser-pulses, the incident direction, the magnitude of the external field and the temperature, the authors prove that the THz emission of KNiF₃/Pt and KCoF₃/Pt originates from the picosecond SSE, and the sd-coupling at the interface plays a key role. The different temperature dependent behaviors of these two samples are explained by the difference between the magnon gap of KNiF₃ and KCoF₃. The experimental results in this paper are relatively clean, and the experimental results are consistent with the physical mechanism given by the authors. The proposed mechanism gives deeper understanding on the picosecond spin transfer process at the interface of AFI/HM and the role of magnetic gap. Therefore, I suggest the publication of this manuscript after the authors address the following minor concerns.

1. There are three kinds of magnetic domains in the sample. When the magnetic field is applied along the [100] direction, the magnon modes in the domain whose Neel vector is along the [100] direction is different from those in the other two domains. All domains contribute to the THz signal. However, the contributions of different domains are not considered in the manuscript. If three domains are taken into account, the authors may make different conclusions.

The Reviewer is correct in stating that all three types of domains, or at least two types^a, will contribute to the THz emission, but this does not affect our conclusions. Although both KNiF₃ and KCoF₃ are cubic antiferromagnets, in the small-oscillation limit they can be approximated to uniaxial antiferromagnets, which have the modes shown in Fig.1 below. In both types of domains, with the Néel vector parallel or perpendicular to the external magnetic field, the $k=0$ magnon modes are characterised by the same energy at zero field, as also confirmed in the case of KNiF₃ by far-infrared absorption spectroscopy¹. In our work we explain the attenuation in the THz emission below T_N in terms of the magnon gap opening at the centre of the Brillouin zone. Because all three types of domains are characterised by the same magnitude of the gap, we expect a similar attenuation of the picosecond spin-Seebeck effect.

For domains not aligned with the field there may also be a contribution to the spin current from coherent spin torques² and the non-linear spin-Seebeck effect³. The coherent spin torques on each sublattice cancel due to symmetry and because the distribution of spin-up and spin-down electrons in the Pt is not significantly altered. The non-linear contribution to the SSE is

^a In KNiF₃ the field of 0.85 T that we apply in the measurement is sufficient to annihilate the [100] domains with the Neel vector parallel to the external field, as detailed in the Supplementary Information. So, in this case the contribution would be mainly from two types of domains, i.e. [010] and [001]. In the case of KCoF₃ the critical field for domain wall motion is higher, about 3.5T, and we expect all three types of domains to be present.

proportional to the induced magnetisation, which is extremely small at the values of magnetic field employed in our experiment.

We have included a discussion on the contribution of the different domains to the THz spin-Seebeck signal in the Supplementary Information, in the paragraph “Magnetic Domains in KCoF_3 and KNiF_3 ”.

2. It is more common to use Pt and W with opposite spin Hall angles to prove that the signal is from the inverse spin Hall effect. Therefore, it is suggested to replace Fig. 2b with the experiments on Pt and W. And in this fig, there should be differences in the positions of peaks for pumping from different side due to the delay of the photon propagation in the samples. However, I find that the THz signals peak at the same time.

The Reviewer is completely right, we indeed observe a shift in the positions of the THz emission when flipping the sample, due to the different lengths of the optical paths involved. For the sake of direct comparability, we plot the time-traces by rescaling the time axis, so that both data sets have the same time axis. We have clarified this in the caption of Fig.2b.

We understand that comparing two metals with opposite spin Hall angle to prove that the origin of the signal resides in the spin Hall effect, is a standard procedure. Nevertheless, we believe that the evidence we have provided is sufficient to fully exclude alternative interpretations. This evidence consists of two main aspects:

1. If the experiment is performed without Pt, i.e. on the bare antiferromagnet, no THz emission is measured. Detecting THz emission requires both a spin source, the antiferromagnet, and a spin-to-charge converter, the Pt. If one of these two elements is absent, we do not observe a THz emission. This is shown in the Supplementary Information.
2. Let us consider the possibility that the deposition of Pt results in additional magnetic phases at the interface contributing to the signal via a magneto-dipole emission. Although this effect, even if present, would produce a very small signal, we can completely exclude it relying on symmetry arguments. When we flip the sample relative

to the pump direction and keep the direction of the external magnetic field unchanged, we would expect the same sign of the emitted THz pulse if its origin laid in a magnetic-dipole. Differently we observe a flipping of the sign in line with the symmetry of the spin Hall effect. This is further explained in the reply to point 3 below.

Additionally, we are unfortunately currently unable to perform new experiments with W. The deposition of W would need to be performed on a virgin specimen, as we are bound to preserve the samples already used for the measurements in this article. Our antiferromagnets were obtained from collaborators in Russia, however transfer of materials has become extremely complex due to sanctions imposed by our funders. Obtaining new specimens to perform the experiments suggested would significantly delay publication.

3. The reason why the signals of KNiF_3 and KCoF_3 are opposite can be explained in more detail in the manuscript. It is not straightforward for the readers.

We agree that this can be confusing due to the different pumping layouts we used for the two materials. We have clarified this point with a new figure of the layouts. Fig.2 below will now become Fig.1a in the new version of the article.

For KNiF_3 we use layout 1 and for KCoF_3 we use layout 2, because these layouts guarantee higher signal-to-noise ratios in both cases. The reason why layout 2 yields a small signal in KNiF_3 is that the pump wavelength (800 nm) coincides with an absorptive window with absorption coefficient of $\alpha=70 \text{ cm}^{-1}$. So, to maximise the absorption of the laser pulse energy by the Pt layer, we need to pump directly on the Pt side.

However, as shown in Fig. 2, in the two pumping layouts the spin current I_{sd} , which always goes from the AFM to the Pt, has opposite directions, while its spin polarisation σ remains pinned along the direction of the external magnetic field H . Because the spin Hall current I_c depends on I_{sd} and σ via the well-known cross product rule $I_c \sim I_{sd} \times \sigma$, a sign change in I_{sd} results in a sign change in I_c , therefore in an opposite sign of the THz emission.

4. The temperature dependence is the key for this work. Do the authors consider the temperature rise caused by the thermal accumulation of the fs laser-pulses irradiation on the sample, particularly for using high frequency laser? How much impact will this have on the results of temperature dependence?

The Reviewer raises a good point concerning the possible temperature drifts induced by illuminating the sample with high-power femtosecond laser pulses. In our experiment this temperature drift is negligible, or below the measurement sensitivity, because of the slow repetition rate of our laser (1 pulse every 200 microseconds) and the relatively low fluences used for the measurements. If a temperature drift took place, this would be similar in the two antiferromagnets because of the similar values of the specific heat and thermal diffusivity⁴. If such drift could affect our data, we would have observed a fluence-dependent shift in the temperature dependence of the spin-Seebeck signal. Instead in KCoF₃ we observe that the transition to a different power law coincides with the Néel temperature at all fluences, as perhaps clearer in the non-normalised data shown in Fig.3.

Reviewer 2

The authors employ pump-probe THz emission spectroscopy in two different antiferromagnetic insulator-heavy metal (AFI/HM) heterostructures, namely KCoF₃/Pt and KNiF₃/Pt. The generated THz signal is attributed to spin current generation by heating the electronic system of Pt with the laser pulse and subsequent spin-charge conversion by the inverse spin Hall effect in Pt. When studying the THz signal in KCoF₃/Pt and KNiF₃/Pt as a function of temperature, the authors find qualitatively different temperature dependencies. For KCoF₃/Pt the THz signal decreases below the Néel temperature T_N and for KNiF₃/Pt the THz signal increases below T_N . Furthermore, the temperature evolution of the THz signal depends on laser fluence in KCoF₃/Pt but not in KNiF₃/Pt. The authors attribute these

experimental findings to the different size of the spin-wave gap in these two antiferromagnetic materials. The observation in KCoF₃/Pt is in qualitative agreement with a theoretical model that employs the presumed large spin-wave gap (>1 THz) in KCoF₃. Spin transport in AFI/HM heterostructures is a relevant topic of current interest due to the rapidly evolving field of antiferromagnetic spintronics. Studies that probe these transport properties at ultrafast timescales are required to obtain a fundamental understanding of the underlying mechanisms. I thus found the study intriguing and timely and it could in principle be suitable for publication in Nature Communications. However, I believe that more detail about the experimental procedure needs to be provided to make the work reproducible and some assumptions warrant additional evidence / explanation. Specifically, these are my questions / requests:

1) The schematic illustration of the experimental setup in Fig. 1 (a) is insufficient. In the text the authors mention that they excite the bilayers either by illuminating from the AFI side or the Pt side. However, the exact geometry, such as the location of the THz detector relative to the sample and pump pulse is not explained clearly. The figure does not even label the AFI or HM, so for the reader it is unclear which geometry was used for the various datasets in Figs 2 & 3. To make the study reproducible please clearly state in the manuscript for each dataset what the measurement geometry was (e.g. Pump-> AFI/Pt -> Detector or Pump & Detector <-> Pt / AFI).

Following the Reviewer's recommendation, we have modified Fig.1a to include a description of the different pumping configurations. In the main text we always specify in which configuration the data were collected.

2) In Fig. 2 (b) was the change of illumination achieved by rotating the sample by 180°? Have the authors ensured that the orientation of the external magnetic field relative to the spin current polarization remained identical? Please explain the argument in line 198 that the relative propagation direction of the THz pulse and spin current changes the sign with a clear figure showing the assumed Néel vector / spin current flow direction / THz polarization.

The two THz pulses in Fig. 2b were measured for the same KCoF₃/Pt sample by using pumping layout 1 (blue curve) and layout 2 (red curve), displayed in Fig.2 (or Fig.1a in the new version of the manuscript). As explained in the answer to point 3 of Reviewer 1, in these two layouts the direction of the spin current I_{sd} is reversed. The polarisation of the spin current, on the other hand, is always along the direction of the magnetic field, which does not change in the two cases, being always parallel to the [100] direction. This is true for both types of domains, with the Néel vector parallel or perpendicular to the field. Because the THz pulse is generated by an impulsive spin Hall current I_c in Pt, a change in sign of I_{sd} results in a change in sign of I_c , thus implying a change in sign of the THz pulse.

3) As explained in the SI, the authors deal with a multi-domain state in KNiF₃/Pt and according to Ref 3 in the SI the Néel vector of the dominant domains will be orthogonal to the external magnetic field. The Model (Eq. (1) in main text) assumes that the external magnetic field is aligned with the Néel vector. The model thus cannot be applied for KNiF₃/Pt with a net magnetic moment stemming from the canting.

As the Reviewer mentions, both KCoF₃ and KNiF₃ are in a multi-domain state during the measurement and all domains contribute to the THz emission. However, as explained in our answer to point 1 of Reviewer 1, this fact does not modify our conclusions. A discussion on

the contribution of the different domains to the THz spin-Seebeck signal is now included in the paragraph “Magnetic domains in KCoF₃ and KNiF₃” of the Supplementary Information.

4) How do the authors ensure that they have a single domain state in KCoF₃/Pt and that the external magnetic field is applied along the Néel vector?

We apologise for the lack of clarity about this point: KCoF₃ is in a multidomain state during our experiments. As we discuss in the Supplementary Information, the maximum field we can apply in our set-up is 0.85 T. Although this is above the critical field for irreversible domain wall motion in KNiF₃ (~0.5 T) it is considerably lower than this critical value for KCoF₃ (~2.9 T). Tanner et al.⁵ showed that when applying an external magnetic field more intense than this critical value domains with the Néel vector parallel to the field shrink, favouring those where the Néel vector lies perpendicular to the field. So, we can safely assume that in KNiF₃ mainly two domains, the [010] and the [001] contribute to the THz emission. On the other hand in KCoF₃ we have to consider contributions from all of the domains, [100], [010] and [001]. As we have specified in our reply to point 1 of Reviewer 1, this does not affect our conclusions.

5) I do not see a strong argument why the different size of the spin-wave gap should be the dominant (sole) cause of the observed different THz emission vs. temperature. Why is the magnitude of the net magnetic moment not relevant? In KNiF₃/Pt, the net moment from canting (see point 3) would lead to a quasi-ferromagnetic spin Seebeck effect increasing with lowering T (as demonstrated by authors in Fig. 3 (d)) while in KCoF₃/Pt (assumed as an ideal uniaxial AFM) the χ_{\parallel} decreases with decreasing temperature resulting in a reduced net moment for lower T (because the net moment stems from thermal fluctuations). A simple model that assumes that the spin-current generation is simply proportional to the net magnetic moment present at the given temperature and magnetic field then would seem to be sufficient to explain the observations. Why is the spin-wave gap model required/better/more plausible?

This is a crucial point, and we thank the Reviewer for this opportunity to further clarify it. Our main conclusion is that the gap in the magnon dispersion plays a key role in the spin Seebeck effect. The argument suggested by the Reviewer, i.e. that the temperature behaviour of the spin Seebeck effect is mainly attributable to the magnetic susceptibility, is indeed valid above T_N . In the paramagnetic phase the spin Seebeck signal cannot be interpreted in terms of magnons, as the long-range magnetic order is absent. However, even in the absence of a spontaneous magnetic moment the magnetic field induces a finite magnetic moment proportional to the magnetic susceptibility. In this case the temperature dependence of the spin Seebeck effect can be entirely described in terms of the critical scaling of the susceptibility as a function of two parameters: the reduced temperature $t = \frac{T-T_N}{T_N}$ and the magnetic field⁶.

The fluence of the pump does not change the scaling law as it has no impact on the magnetic ordering or the susceptibility. Effects connected to accumulated heating can be excluded as argued in details in our reply to point 4 of Reviewer 1.

As expected, Fig. 4(a) shows that the temperature dependence of the picosecond spin Seebeck signal in KCoF_3 does not depend on the fluence at $T > T_N$. Moreover, this temperature dependence overlaps with the temperature dependence of the spin Seebeck effect measured in the quasi-dc limit and in a completely different antiferromagnet, FeF_2 ⁷. We have extended the section “THz emission in the paramagnetic phase of KCoF_3 ($T > T_N$)” of the Supplementary Information and added the graph shown in Fig. 4(a) of this reply.

The description of the spin Seebeck effect in terms of a critical phenomenon relies on the assumption that the free energy is a homogeneous function of two variables only: the reduced temperature and the magnetic field⁶. Below T_N this assumption collapses as we have measured a clear dependence on the fluence. This observation implies that the thermodynamic quantities describing the system at equilibrium, such as the susceptibility, are not sufficient to capture the phenomenology of the transient states. We thus must also consider the energetics of the process. In the model that we propose the efficiency of spin generation/transfer across the interface depends on the overlap between the magnon density of states and the Fermi distribution function of the free electrons in Pt. The presence of the magnon gap hence strongly suppresses the spin transfer efficiency, when its magnitude is higher or comparable with the energy gain of the free electrons in Pt. This energy gain is roughly 6 meV^3 for a pump fluence of 0.1 mJ/cm^2 . We therefore expect a significant attenuation of the spin Seebeck effect in correspondence of the lowest values of fluence employed in the experiments.

In the case of KNiF_3 , even at the lowest fluences used in the experiment the magnitude of the magnon gap is 5 times lower than the energy gain of the free electrons and its contribution is thus negligible. Interestingly, as shown in Fig. 3d of the main text, in this case we observe a critical-like behaviour: the temperature dependence of the picosecond spin-Seebeck effect in KNiF_3 does not depend on the pump fluence and overlaps with that of antiferromagnetic MnF_2 ⁸, which also has a small gap of 1.08 meV , and that of ferrimagnetic YIG ^{9,10}, plotted as a function of the reduced temperature (Fig. 4(b)). To the best of our knowledge no satisfactory explanation was given for the exponent 3, which is not related to the critical exponent for the magnetization, nor for the magnetic susceptibility.

6) The fluence dependence below T_N is much stronger in experiment (Fig 3(a)) than in theory (Fig. 3(c)). Why?

In the model presented in the manuscript there was a theory-experiment mismatch over the interpretation of ΔT_e (whether it was the peak temperature, or the total temperature increase). Hence the results were over-conservative about the peak electronic temperature in the Pt in the experiments and in the model. We have now recalculated the results assuming that the peak electronic temperature increases linearly with the pump fluence. We have found a very good agreement between the experiment and the model. Also noting the change to the τ_e parameter as discussed in the reply to Reviewer 3, point 5.

Reviewer 3:

The authors describe their experiment and modeling of the generation of THz radiation by the so-called picosecond spin Seebeck effect created at an interface between a heavy metal and an insulating antiferromagnet in an applied magnetic field. The picosecond spin Seebeck effect was first identified in interfaces with insulating ferromagnets in 2017. This work extends the scope of these types of investigations to insulating antiferromagnets. The work is timely and important. I recommend publication in Nature Communications after the authors have considered the following criticisms:

1) The magnetic susceptibility of $KCoF_3$ has a much weaker temperature dependence below T_N than $KNiF_3$. The susceptibility of $KNiFe_3$ behaves approximately as I would expect but $KCoF_3$ does not. In fact, since the magnon gap is expected to be 1 THz, i.e., 4 meV, I would expect that the magnetic susceptibility would fall off more steeply with decreasing temperature than $KNiF_3$ as the thermal excitations of magnons is quenched at low temperatures. The authors should discuss this point and provide a reasonable explanation of the observed magnetic susceptibility of $KCoF_3$.

Indeed, the unusual susceptibility curve of $KCoF_3$ has been known since the 70's^{11,12,13}. In the literature the high value of the longitudinal susceptibility at low temperatures is attributed to the residual magnetic moment originating in the unquenched orbital angular momentum of the Co^{2+} atoms. Moreover, strain is responsible for domain rearrangement, favouring those domains with the Néel vector parallel to the stress direction¹⁴. Our susceptibility curve shown in Fig.1c of the main text is very similar to the susceptibility reported in [12] (Fig. 5) when the $KCoF_3$ specimen is subjected to a perpendicular strain. Although in our measurements we don't actively apply a stress to our sample, a strain can originate from the way the sample is mounted and glued. Although we have not carried a systematic study on strain, we have observed that the magnetic susceptibility curves measured on the same $KCoF_3$ specimen can vary significantly, if the sample is removed and remounted in the set-up. We provide an example below. Here the susceptibility is measured along the [001] crystal direction and no magnetic field is applied during the cool down. The higher susceptibility in run 1 suggests a larger contribution from the domains with the Néel vector oriented in the (001) plane.

We have added this discussion and Fig. 5 in the section “Magnetic Domains in KCoF₃ and KNiF₃” in the Supplementary Information.

2) In Figure 2, the authors plot the data normalized by the largest value of “peak S(t)” as a function of temperature. I find this practice misleading and it hides much of what I would like to compare in the data. I strongly recommend that the authors plot the observed (not normalized) “peak S(t)” with the addition perhaps of dividing by the pump fluence when comparing data obtained at different pump fluences.

We think the Reviewer is referring to the temperature dependence of the spin-Seebeck effect in Fig.3 rather than Fig.2 of the main text. Plotting the renormalized data has the specific purpose of rescaling by the value of ΔT_e , the increase of the effective electronic temperature, and access the temperature dependence of the spin-mixing conductance. The surface contribution of the spin Seebeck effect that we are probing in this experiment is described as¹⁵

$$j_{sd} = \frac{\gamma \hbar k_B g^{\uparrow\downarrow}}{2\pi M_s V} \Delta T_e.$$

We know that ΔT_e depends on the fluence but we do not know the exact relation between the two parameters (the values that we provide for the absorbed fluence are estimates based on a continuous rather than an impulsive wave). Referring to our reply to point 5 of Reviewer 2, above the Néel temperature the spin-Seebeck signal is described by $f(t,H) \Delta T_e$, where $f(t,H)$ is a function of the reduced ambient temperature and magnetic field and does not depend on the fluence. Moreover, ΔT_e does not depend on ambient temperature¹⁶. Therefore, renormalising the data taken at a specific fluence φ_0 by $\Delta T_e(\varphi_0)$ is achieved by dividing the original signal by the value of the spin Seebeck signal at that fluence and at a fixed temperature above T_N , $f(t_0, H) \Delta T_e(\varphi_0)$.

If we follow this normalisation procedure, we obtain a fluence-independent function above the Néel temperature, as expected. Below the Néel temperature however we do not: this indicates

that the spin mixing conductance does not only depend on temperature and magnetic field but also on fluence. This conclusion is a central result of our work.

In any case, we agree with the Reviewer that providing the original unnormalised data is useful and we have therefore added the graph in Fig.6 in the Supplementary Information.

3) One of the questions I would like know the answer to (related to the point above) is the magnitude of the signals for YIG/Pt and KNiF₃/Pt. The magnetization of KNiF₃ in the approximately 1 T field is orders of magnitude smaller than the magnetization of YIG. Is the THz generation also orders of magnitude smaller? If not, the authors should discuss the relative magnitudes of the signal for these two cases.

We agree with the Reviewer that this information should be present in the paper and we have included the graph in Fig. 7 in the Supplementary Information. It is established by theory¹⁷ and experiment⁴ that the paramagnetic spin Seebeck effect and the antiferromagnetic spin Seebeck effect at temperatures $\geq T_N$ are proportional to the external magnetic field times the spin susceptibility of the material. We therefore expect the amplitude of the THz emission to be more than ten times larger in KCoF₃ than in KNiF₃ at their respective Néel temperatures. This is not true in the ordered phase of antiferromagnets, as also discussed in reply to point 5 of Reviewer 2. Even in ferromagnetic YIG there is no proportionality of the spin-Seebeck signal with the magnetic moment, which follows the Bloch law, as established in quasi-dc¹⁸ and THz¹⁶ studies. As already discussed in reply to point 5 of Reviewer 2, the spin Seebeck effect in antiferromagnets like MnF₂ and KNiF₃ is described by the same scaling law, but to the best of our knowledge, no theory exists that can explain it. Magnitude-wise we note that YIG has a magnetic moment orders of magnitude larger than antiferromagnets under the action of a 1 T field. In spite of this fact we experimentally establish that the amplitudes of the spin-Seebeck effect in YIG and KCoF₃ are comparable, further confirming that the key parameter is definitely not the magnetisation. The understanding of the spin Seebeck effect in heterostructures is nevertheless still at the beginning and thus further experimental and theoretical studies are required.

4) The authors must provide a thorough discussion of the preparation and surface characterization of the KCoF₃ and KNiF₃ crystals. In this version of the manuscript, all that is stated is that the crystals are polished to a roughness of 1 nm (which also seems to me to be extraordinarily small roughness for a mechanically polished crystal). Presumably the crystals are covered with adsorbed water vapor and hydrocarbons before loading into the deposition chamber. How are these surface contaminants removed prior to Pt deposition? This work is about interface physics and the structure and composition of the interfaces is important.

The crystals were polished using a Kent 3B automatic fibre optic polishing machine. We first coarse-polished about 100 microns of material by using a 15 μm diamond suspension on planacloth. Then, we gradually decreased the abrasive size from 15 μm to 9, 6, and 3 μm . Between two stages, the sample was cleaned with de-ionized water to remove residues from the previous polishing stage. For the fine polishing 1 μm of alumina silica was used on multicloth. De-ionised water is constantly added during the polishing to avoid the cloth getting dry. In the final stage, we polished the sample with 0.06 μm alumina silica on multicloth and cleaned the sample with acetone and isopropanol. In Fig. 8 we report two sample AFM scans for KNiF_3 and KCoF_3 . The achieved roughness is about 7 nm in KNiF_3 and 3 nm in KCoF_3 .

In addition to polishing, the surface was further cleaned by in-situ Ar-milling at the pressure of 2.2×10^{-4} mbar (base pressure is 3.5×10^{-8} mbar) before sputtering the Pt.

This description has now been included in the methods.

5) The authors state the values of the model parameters at the bottom of page 12 without discussing if these values are physically reasonable. For example, is it reasonable for J_{SD} to be 5 times larger than J_{AFI} ? The text states that J_{SD} is considered a small parameter yet its value is larger than J_{AFI} . The text states that the electron-phonon relaxation time in Pt is 0.2 ps but that value is given as 2 ps for the model.

Regarding J_{sd} , we argue that J_{sd} is not dominant in the total Hamiltonian, rather than stating that J_{sd} is small in comparison with J_{AFM} . Hence H_{sd} is treated as a perturbation of the system so that the eigenmodes of the AFI and NM are not significantly altered by the presence of H_{sd} . Note that the AFI and NM are modelled as bulk-like systems (as is standard in similar analytic approaches), whereas J_{sd} is a coupling at a single plane in space. Hence even if J_{sd} is very large, the effect on the bulk AFI and NM eigenmodes will be very small. Related to this point, J_{sd} is a parameter which seems to be poorly quantified in the literature. Different sources use values in the range 1-0.01 eV, while still considering H_{sd} not to dominate the Hamiltonian.

The Reviewer is correct about the discrepancy between 0.26 ps and 2 ps for the electron-phonon relaxation time. We thank her/him for pointing out the mistake. Consequently we have recalculated the model results using 0.26 ps. The peak electronic temperature is slightly changed, but the general spin current results remain approximately the same. This is because the speed of the spin current dynamics was already dominated by the much faster Pt spin-flip time, hence the longer thermal dynamics given by using 2 ps for the electron-phonon timescale were of little impact.

The general critique of the Reviewer about what can be considered as reasonable model values is very valid and has enabled us to identify an error in our derivation. We identified two missing volume factors (originating from the volume of the Wigner-Seitz cell of the AFI and NM when converting a sum over reciprocal space into an integral). We now use values for J_{sd} and electronic density of states at the Fermi energy from Ref.[¹⁹], in which spin currents in NiO/Pt layers were studied both theoretically and experimentally in the DC limit. All other parameters relate to measured quantities with appropriate citations. After recalculating, our results are qualitatively unchanged. Correcting the missing volume factors compensated our arbitrary choices of J_{sd} and D , so the spin current pre-factor remained of the same order of magnitude.

6) At the end of the paper the authors state: “Our results set a landmark in the fundamental understanding...”. I understand that many authors feel a need to hype their work to publish in so-called “high impact journals”. The authors describe a thorough and important contribution to the literature of spin current generation and transport at interfaces. That is enough. Calling this work a “landmark” study is going too far.

Implemented, as suggested.

-
- ¹ H. Yamaguchi et al., *Antiferromagnetic resonance in the cubic perovskite KNiF₃*, Phys. Rev. B **59**, 6021 (1999)
- ² R. Cheng et al., *Spin Pumping and Spin-Transfer Torques in Antiferromagnets*, Phys. Rev. Lett. **113**, 057601 (2014)
- ³ T. S. Seifert et al., *Femtosecond formation dynamics of the spin Seebeck effect revealed by terahertz spectroscopy*, Nat. Commun. **9**, 2899 (2018)
- ⁴ A. Oleaga et al., *Critical behaviour of magnetic transitions in KCoF₃ and KNiF₃ perovskites*, Journal of Alloys and Compounds **629**, 178 (2015)
- ⁵ Safa, M. & Tanner, B. K. Antiferromagnetic domain wall motion in KNiF₃ and KCoF₃ observed by X-ray synchrotron *topography*, Philosophical Magazine B **37**, 739–750 (1978)
- ⁶ M. D. Vannette et al., *Precise measurements of radio-frequency magnetic susceptibility in ferromagnetic and antiferromagnetic materials*, Journal of Magnetism and Magnetic Materials **320**, 354 (2008)
- ⁷ J. Li et al., *Spin Seebeck Effect from Antiferromagnetic Magnons and Critical Spin Fluctuations in Epitaxial FeF₂ Films*, Phys. Rev. Lett. **122**, 217204 (2019)
- ⁸ S. M. Wu et al., *Antiferromagnetic spin Seebeck effect*, Phys. Rev. Lett. **116**, 097204 (2016)
- ⁹ F.N. Kholid et al., *Temperature Dependence of the Picosecond Spin-Seebeck Effect*, Applied Phys. Lett. **119**, 032401 (2021)
- ¹⁰ K. Uchida et al., *Quantitative Temperature Dependence of Longitudinal Spin Seebeck Effect at High Temperatures*, Phys. Rev X **4**, 041023 (2014)
- ¹¹ K. Hirakawa et al., *Magnetic Properties of Potassium Iron Group Fluorides KMF₃*, Journal of the Physical Society of Japan **15**, 2063 (1960)
- ¹² Tsuda et al., *Magnetic Susceptibility of an Antiferromagnetic KCoF₃ Single Crystal—¹⁹F NMR and Static Measurements*, J. Phys. Soc. Jpn. **45**, 1551 (1978)
- ¹³ N. Suzuki et al., *Theoretical study of magnetic susceptibility of orbitally unquenched compound KCoF₃*, Solid State Communications **23**, 319 (1977)
- ¹⁴ J. Ferre, *Magnetic-field- and stress-induced linear birefringence in cubic antiferromagnets RbMnF₃, KNiF₃ and KCoF₃*, J. Phys. C: Solid State Phys. **16** 3971 (1983)
- ¹⁵ Xiao, J., Bauer, G. E. W., Uchida, K. C., Saitoh, E. & Maekawa, S. *Theory of magnon-driven spin Seebeck effect*, Physical Review B **81**, 214418 (2010)
- ¹⁶ F.N. Kholid et al., *Temperature Dependence of the Picosecond Spin-Seebeck Effect*, Applied Phys. Lett. **119**, 032401 (2021)
- ¹⁷ Y. Yamamoto et al., *Spin Seebeck effect in paramagnets and antiferromagnets at elevated temperatures*, Phys. Rev. B **100**, 064419 (2019)
- ¹⁸ T. Kikkawa et al., *Critical suppression of spin Seebeck effect by magnetic field*, Phys. Rev. B **92**, 064413 (2015)
- ¹⁹ K. Chen et al., *Temperature dependence of angular momentum transport across interfaces*, Phys. Rev. B **94**, 054413 (2016)

Reviewers' Comments:

Reviewer #1:

Remarks to the Author:

The authors revised the manuscript with additional explanations. All my issues are properly addressed, and the paper is more much convincing. Now I recommend the publication of the present manuscript.

Reviewer #2:

Remarks to the Author:

The authors have replied to all questions of all three reviewers in detail. I found their arguments persuasive and the revised manuscript is now sufficiently clear in experimental detail for the study to be reproducible. The revised manuscript and SI also successfully address my further points of criticism from the first review round.

All in all, the review process has very clearly improved the manuscript and I have no further reservations or remarks. The topic remains relevant and timely as already pointed out in my initial review. I thus recommend publication of the revised manuscript in Nature Communications.

Reviewer #3:

Remarks to the Author:

The authors have adequately addressed the concerns raised in my first report. I recommend publication in Nature Communications.